# Intranasal Delivery of Mesenchymal Stromal Cells Protects against Neonatal Hypoxic–Ischemic Brain Injury

**DOI:** 10.3390/ijms20102449

**Published:** 2019-05-17

**Authors:** Courtney A. McDonald, Zlatikha Djuliannisaa, Maria Petraki, Madison C. B. Paton, Tayla R. Penny, Amy E. Sutherland, Margie Castillo-Melendez, Iona Novak, Graham Jenkin, Michael C. Fahey, Suzanne L. Miller

**Affiliations:** 1The Ritchie Centre, Hudson Institute of Medical Research, Clayton 3168, Australia; d.zlatikha@gmail.com (Z.D.); mpet44@student.monash.edu (M.P.); madison.paton@cerebralpalsy.org.au (M.C.B.P.); trpen2@student.monash.edu (T.R.P.); amy.sutherland@hudson.org.au (A.E.S.); margie.castillo-melendez@hudson.org.au (M.C.-M.); graham.jenkin@monash.edu (G.J.); suzie.miller@monash.edu (S.L.M.); 2Department of Obstetrics and Gynaecology, Monash University, Clayton 3168, Australia; 3Cerebral Palsy Alliance Research Institute, University of Sydney, Camperdown 2086, Australia; INovak@cerebralpalsy.org.au; 4Department of Paediatrics, Monash University, Clayton 3168, Australia; michael.fahey@monash.edu

**Keywords:** cerebral palsy, cord tissue cells, mesenchymal stem cells, neurodevelopment, perinatal brain injury

## Abstract

Cerebral palsy (CP) is a permanent motor disorder that results from brain injury and neuroinflammation during the perinatal period. Mesenchymal stromal cells (MSCs) have been explored as a therapy in multiple adult neuroinflammatory conditions. Our study examined the therapeutic benefits of intranasal delivery of human umbilical cord tissue (UC) derived-MSCs in a rat model of neonatal hypoxic–ischemic (HI) brain injury. To do this, HI was performed on postnatal day 10 Sprague-Dawley rat pups via permanent ligation of the left carotid artery, followed by a hypoxic challenge of 8% oxygen for 90 min. A total of 200,000 UC-MSCs (10 million/kg) were administered intranasally 24 h post-HI. Motor control was assessed after seven days, followed by post-mortem. Analysis included brain immunohistochemistry, gene analysis and serum cytokine measurement. Neonatal HI resulted in brain injury with significant loss of neurons, particularly in the hippocampus. Intranasal administration of UC-MSCs significantly reduced the loss of brain tissue and increased the number of hippocampal neurons. HI significantly upregulated brain inflammation and expression of pro-inflammatory cytokines, while intranasal UC-MSCs significantly reduced markers of neuroinflammation. This study demonstrated that a clinically relevant dose (10 million/kg) of UC-MSCs was neuroprotective following HI by restoring neuronal cell numbers and reducing brain inflammation. Therefore, intranasal delivery of UC-MSCs may be an effective therapy for neonatal brain injury.

## 1. Introduction

Cerebral palsy (CP) is the most common cause of physical disability in childhood [1]. In developed countries, 1 in 400 babies born will develop CP, and these figures are even higher in developing countries. This creates a large financial burden for families and the healthcare system, with the lifetime costs for a child with CP estimated at just under US$1 million per individual [1]. Current therapeutic options for CP are limited; however, there are a number of clinical trials underway investigating stem cells as regenerative treatments for children with established CP [2]. CP is caused by damage to the developing brain, predominantly during pregnancy or at birth, and hypoxia–ischemia (HI) at term birth is a readily identifiable cause of neonatal brain injury and subsequent CP. The only current therapy for term HI is therapeutic hypothermia that must begin within 6 h after birth. While this is effective in some neonates, more than half of those treated still die or suffer severe neurological damage [3]. There is, therefore, a need for novel therapies that can be used as alternative or complimentary therapies with hypothermia. One such potential therapy is the use of mesenchymal stromal cells (MSCs).

MSCs can be readily sourced from bone marrow, adipose and placental tissues, particularly the umbilical cord (UC) [4], and many private umbilical cord blood banks now routinely collect and store UC tissue for future MSC isolation. MSCs are the most widely studied cell type of any stem cell, with over 400 currently registered clinical trials using adult-derived MSCs for a variety of conditions, including neurological dysfunctions [5]. Due to greater telomerase activity, UC-derived MSCs are more clonogenic and demonstrate a higher proliferation capacity than adult MSCs [6]. Importantly, UC-derived MSCs are also reported to have a greater neurotrophic effect than bone marrow MSCs [7]. The neuroprotective and reparative capacity of MSCs is primarily facilitated by their immunomodulatory properties and their capacity to secrete neurotrophic and anti-apoptotic factors [4,8]. We and others have shown that the immunomodulatory function of MSCs is multifaceted, demonstrating not only the direct secretion of anti-inflammatory cytokines but also altering immune cell programming and cell proliferation [9,10,11]. MSCs can modify dendritic cells and the release of pro-inflammatory cytokines, such as tumour necrosis factor (TNF), from macrophages, [10]. These actions convey long-term systemic immune inhibition, which could be protective in the setting of hypoxia–ischemia-induced neuroinflammation and perinatal brain injury.

The potential of MSCs for the treatment of neonatal brain injury has been previously reported using systemic (i.e., intravenous) delivery of cells [11,12]. However, it is apparent that most systemically administered cells become trapped in the lungs, partly due to the size of the cell [13]. In light of this, examination of the neuroprotective benefits of MSCs should consider other routes of administration. Indeed, previous studies have examined the neuroprotective actions of intranasally administered MSCs for neonatal brain injury, using very high doses of cells (50–200 million cells/kg). By contrast, a clinical trial of MSCs for lung injury (bronchopulmonary dysplasia) in a paediatric population used 10 million cells per kg delivered via an intratracheal route [14] and all other trials using intravenous MSC administration to paediatric populations are using lower cell doses. Thus, there is a discrepancy in MSC dosing between preclinical studies which demonstrate efficacy and clinical trials.

Here we adapted a well-described rat model of HI brain injury that mimics human term-born hypoxic–ischemic brain injury [15] and used it to assess the therapeutic potential of a clinically relevant dose of UC-MSCs via the intranasal route at 24 h post-HI, for the treatment of neonatal HI brain injury. Our aim was to examine whether intranasal UC-MSCs reduced (i) short-term motor control deficits; (ii) neuroinflammation (microglial activation and astrogliosis); and (iii) neuronal cell death and tissue loss, induced via neonatal HI. We hypothesised that a clinically relevant dose (10 million/kg) of UC-MSCs, delivered intranasally, would be an effective treatment strategy for neonatal HI brain injury, mediated via their anti-inflammatory/immunomodulatory actions.

## 2. Results

### 2.1. The Effect of Intranasal Umbilical Cord Mesenchymal Stromal Cells (UC-MSC) Treatment on Behavioural Outcomes

Behavioural assessment for motor strength and control was examined using the negative geotaxis test performed 4 days after HI (PND14). Two parameters were measured, the time taken for the rats to turn 180° (Figure 1A), and the time taken for rats to walk up the board and cross a designated line (15 cm; Figure 1B). The time taken for HI rats to turn 180° was significantly increased compared to sham (*p* < 0.01; Figure 1A), while UC-MSC treatment significantly improved this deficit (*p* < 0.05), and UC-MSC was not different to sham. There were no significant differences in the time taken to walk up the board in any group; however, there did appear to be a trend for an increase in time in the HI group, and return to sham-equivalent in the intranasal UC-MSC treatment group, as seen in Figure 1B.

### 2.2. The Effect of Intranasal UC-MSC Treatment on Hypoxic–Ischemic (HI) Brain Injury

At the time of post-mortem 7 days after HI, body weights were not significantly different between groups (Sham 30.1 ± 1.9 g, HI 26.7 ± 2.3 g, UC-MSC 30.7 ± 1.4 g). However, brain weight was significantly reduced in HI-injured brains (*p* < 0.001; Figure 2A) compared to the sham group. Intranasal treatment with UC-MSCs partially restored brain weight compared to HI (*p* < 0.05) but was still significantly reduced compared to sham (*p* < 0.05; Figure 2A). Given HI injury reduced the overall weight of the brain, we wanted to examine more closely the degree of cerebral tissue loss associated with acute HI in the left brain hemisphere, and observed a significant increase in tissue loss in the HI group compared to sham (*p* < 0.05; Figure 2B). UC-MSC treatment partially rescued left hemisphere tissue loss, with UC-MSC significantly different to HI-injured rats (*p* < 0.05; Figure 2B), but the UC-MSC group also remained significantly increased compared to sham (*p* < 0.05).

We further assessed the number of neurons present to determine if HI injury reduced the number of neurons in two regions of interest, the hippocampus and the somatosensory cortex. We found a significant reduction in the number of neurons in the HI group compared to sham in the hippocampus (*p* < 0.001; Figure 3A), which was significantly improved by intranasal treatment with UC-MSCs (*p* < 0.05; Figure 3B–D). In the cortex, we also observed a significant decrease in the number of neurons in the HI group compared to sham (*p* < 0.01; Figure 3E). UC-MSC treatment did not significantly increase neuron cell counts compared to HI but was not significantly reduced from sham.

### 2.3. The Effect of Intranasal UC-MSC Treatment on Neuroinflammation

To determine if intranasal UC-MSC treatment modified the neuroinflammatory response associated with neonatal HI brain injury, we investigated the activation of microglia and astrocytes. Analysis of microglia in the hippocampus showed a significant increase in the number of activated microglia (*p* < 0.01; Figure 4A), resting microglia (*p* < 0.05; Figure 4B) and total microglia (*p* < 0.001; Figure 4C) in the HI group compared to sham. Intranasal UC-MSC treatment significantly decreased the number of activated microglia (*p* < 0.05) and total microglia (*p* < 0.05) compared to the HI group, but this reduction in microgliosis was only partial when compared to the sham animals (Figure 4B,C). In the somatosensory cortex, no significant differences were observed between any groups in activated (Figure 4G), resting (Figure 4H) or total microglia (Figure 4I).

Analysis of astrocyte activation in the hippocampus illustrated a significant increase in the density of astrocytes in the HI group compared to sham (*p* < 0.001; Figure 5A). Furthermore, intranasal UC-MSC treatment significantly reduced astrocyte density compared to the HI group (*p* < 0.01) to levels similar to the sham group. Results from the analysis of glial fibrillary acidic protein (GFAP) staining in the somatosensory cortex (Figure 5B) found no significant differences in astrocyte density between any groups. Representative images of GFAP staining in the hippocampus are shown for sham (Figure 5D), HI (Figure 5E) and UC-MSC treatment (Figure 5F).

### 2.4. The Effect of Intranasal UC-MSC Treatment on Peripheral Cytokine Expression

We examined whether intranasal UC-MSC treatment modified cytokine concentration in the peripheral circulation. To do this, we measured cytokine protein expression in serum collected at post-mortem. We observed a trend towards an increase in anti-inflammatory Interleukin (IL)-10 concentration in the serum of UC-MSC treated rats compared to the HI group (*p* = 0.06; Figure 6A), but no differences in monocyte chemoattractant protein (MCP)-1 (Figure 6B) or IL-18 (Figure 6C) were observed.

### 2.5. The Effect of Intranasal UC-MSC Treatment on Gene Expression in the Injured Brain

To more closely examine the neuroprotective mechanisms of action of intranasal UC-MSC treatment, we assessed gene expression in the injured left hemisphere, 7 days post-HI. We assessed several inflammatory, neurotrophic, angiogenic and growth factor genes (Table 1). HI injury resulted in a significant decrease in the expression of brain-derived neurotrophic factor (*BDNF*) compared to sham (*p* < 0.05; Table 1), which was not improved with intranasal treatment with UC-MSCs. While not significant, there was a trend towards a decrease in vascular endothelial growth factor (*VEGF*) in the HI group compared to sham, which was not improved with UC-MSC treatment. Similarly, there was a trend towards an increase in insulin-like growth factor-1 (*IGF-1*) following HI, which UC-MSC treatment did not improve. There were no differences observed for all other genes tested, which included *claudin-5*, *occludin* and glial-derived neurotrophic factor (*GDNF*).

## 3. Discussion

Injury to the developing brain in response to acute HI can have serious lifelong consequences, resulting in neurological deficits and conditions such as CP. Currently, term-born infants diagnosed with neonatal encephalopathy receive hypothermia therapy as a neuroprotective intervention, but cooling is only offered to infants born at term and must commence within 6 h after birth to have any effect [16]. In this study, we assessed the ability of human UC-MSCs to reduce neonatal brain injury when given intranasally, 24 h after acute HI insult. Our results showed that intranasal UC-MSCs partially prevent the loss of brain tissue following HI and prevent neuronal loss within the hippocampus. Further, we demonstrated that this neuroprotective action of UC-MSCs was likely mediated via an anti-inflammatory effect as evidenced by a brain region-specific decrease in microglia and astrocyte activation, and increased circulating expression of the anti-inflammatory cytokine IL-10. We did not find that UC-MSCs engrafted within the brain or modified cerebral production of neurotrophic factors such as *BDNF*. These results demonstrate that neuronal cells of the hippocampus are particularly vulnerable to term-equivalent HI, but that intranasal administration of a clinically relevant dose (10 million/kg) of UC-MSCs is neuroprotective, reducing neuroinflammation and neuronal cell loss.

MSCs are the most well-studied cell type for stem/ progenitor cell therapy, with over 400 registered clinical trials investigating the potential of these cells [5]. It is, however, interesting to note that very few MSC products have progressed from the preclinical and clinical trial pipeline to clinical use. One reason for this failure to cross the “valley of death” and translate findings into clinical therapy is the lack of critical information regarding effective cell dosing. Most previous studies have reported MSC efficacy for neonatal brain injury using high doses (>50 million cells/kg) of MSCs [17,18,19,20], which are potentially not feasible due to manufacturing limitations, and lack detailed safety data for neonates and infants. While these studies are valuable and have laid the foundation for the use of MSCs for neonatal brain injury, it is imperative to test the efficacy of clinically feasible and relevant lower cell doses. A recent meta-analysis investigated the effectiveness of MSCs for neonatal HI brain injury in preclinical studies [21]. This study concluded that research into dosing needed further investigation and comparative examination of cell dosages based on body weight rather than a total cell number [21]. Here, we show that a clinically relevant dose of UC-MSCs, equivalent to 10 million cells per kg of body weight, is effective at improving motor behavioural outcomes, reducing neuroinflammation and protects neurons in critical regions of the brain such as the hippocampus. This study is the first to show that a low dose of UC-MSCs given via the intranasal route is neuroprotective for the developing brain.

It is well understood that HI brain injury around the time of birth can lead to the development of CP, and patients often have cognitive impairments, including memory difficulties [22]. It has also been shown that therapeutic hypothermia, the only current therapy for neonatal encephalopathy induced via HI, does not reduce cognitive impairments in school-age children [23]. Moreover, a recent study has shown that therapeutic hypothermia does not protect the hippocampus following neonatal HI in a murine model [24]. Given the crucial role of the hippocampus in memory processing, converting short-term to long-term memories, neuroprotective therapies that target this region are essential. In this current study, we confirm that the hippocampus is particularly vulnerable to neonatal HI brain injury, mediated through significant neuroinflammation. We also demonstrate that intranasal administration of UC-MSCs significantly reduces neuroinflammation and protects hippocampal neurons. In this study, we administered UC-MSCs at 24 h post-HI, thereby extending the window of opportunity for treating with hypothermia. We did not include a group with hypothermia plus UC-MSCs, but given that MSCs may show additional benefits for motor and cognitive functions, further studies with these combined treatments are certainly warranted.

Our study showed that intranasal delivery of UC-MSCs could significantly improve short-term behavioural deficits. Given the young age of the pups used in this study, PND 10-17, there are limited behavioural tests that can be performed. At this age, pups are not very active and have not yet developed innate behaviours, such as rearing; therefore, it is not possible to perform tests such as novel object recognition, forelimb preference tests or open field tests, as done in older rats [25]. The negative geotaxis test can be used in this age group, and it involves assessing the vestibular reflex, strength and coordination. We confirmed in our study that the ability of rats to right themselves and turn 180° after being placed facing down is impaired with HI, and UC-MSCs could significantly improve this deficit. However, we did not observe a significant deficit in the time taken to walk up the board in the HI group, which mainly tests strength and coordination. While these tests are useful to get an understanding about the impact of injury and treatment on motor control, they are not always the best method. It is critical that future studies include long-term behavioural studies that will enable other more advanced behavioural testing to be performed. This will also provide evidence for the long-term efficacy of cell treatment which is very important, especially since we have recently published that assessment of short-term efficacy does not always reflect long-term improvements [25].

As expected based on previous studies [12,14], we demonstrated that intranasal delivery of UC-MSCs reduced neuroinflammation, specifically reactive astrocytes and microglia in the hippocampus region of the brain [17,19]. In this study, we assessed microglial activation using Iba-1 immunohistochemistry and morphological assessment; this method does have limitations. It is therefore important that future studies assess microglial action state more thoroughly to understand how UC-MSCs are affecting these populations. We also showed modulation of the systemic immune response with intranasal delivery of UC-MSCs through increased concentration of the anti-inflammatory cytokine IL-10 in serum, which may also contribute to the neuroprotection observed. From previous studies investigating the mechanisms of cell therapies [26], we know that modulation of the peripheral immune response plays an important role in reducing neuroinflammation, potentially through reduced trafficking of key peripheral inflammatory cells, such as T-cells, that contribute to activation of neuroinflammation and brain injury [10,26,27]. The recruitment of monocytes to the brain during injury is another mechanism that may play an important role in the development of injury. We have previously shown in a sheep model of preterm inflammation-induced brain injury that UC-MSC treatment can reduce the recruitment of monocytes to the brain following injury [28].

A significant decrease in the neurotrophic factor *BDNF* was observed following neonatal HI brain injury but, surprisingly, intranasal UC-MSC treatment did not improve expression of this neurotrophic factor. It has previously been shown that intravenous MSC therapy significantly increases the expression of *BDNF*, and this has been suggested as one of the potential repair mechanisms [17,29]. However, it has not specifically been investigated after intranasal delivery. Whether this reflects an actual difference in the two administration routes, or time between MSC administration and tissue collected, needs to be investigated further. However, we noted a trend towards increased *GDNF* with UC-MSCs, which is another important neurotrophic factor, indicative that this route may induce endogenous neurotrophin release.

Our study aimed to investigate a safe alternative to systemic cell administration that places cells adjacent to the site of damage. It is accepted that very few cells reach the brain following systemic administration [13,30] but, importantly, MSCs are not required to reach the brain for effective neuroprotection with their actions conveyed via paracrine factors [9,11]. There is discussion in the field that if cells could be delivered closer to the brain following injury, their close proximity may improve the release of local cerebral trophic factors and may improve neuroprotection [31]. Unfortunately, routes that deliver cells directly to the brain, such as intracerebroventricular, intraparenchymal or intrathecal injections, possess high risks [31]. By contrast, intranasal delivery of cells is low-risk and is more clinically feasible. The results of this study support previous findings that intranasal delivery of UC-MSCs is effective for treatment of neonatal brain injury [18,20], even though cells used in our study could not be found in the brain at the time of post-mortem. While no direct comparisons between intravenous and intranasal delivery have been performed for neonatal brain injury, in a multiple sclerosis model, intranasal and systemic (intraperitoneal) delivery of MSCs were compared, and intranasal delivery was more effective at reducing injury progression compared to the same dose given systemically [32]. Together, the evidence suggests that, for neurological conditions, intranasal delivery is a valid route that is safer, with fewer complications compared to other administration routes.

Lastly, it is important to note that in our study, the anti-inflammatory and neuroprotective actions of UC-MSCs were partial, and cell therapy did not entirely prevent the progression of brain injury following HI. Further, the effects we observed were brain region specific. It would be of great interest in future studies to examine whether complete neuroprotection could be conferred with adjuvant therapies. For example, co-administration with other cell types, such as umbilical cord blood cells, neural stem cells, or small molecule drugs that may directly target the immune system could be combined with UC-MSCs administration to offer a superior combined therapy. In addition, hypothermia commencing at 6 h followed by UC-MSCs should be examined. We also only studied administration of UC-MSCs at one time point, 24 h post-injury, and while this extends the current treatment window, it would be interesting to test cell administration at later time points and to examine the effects of multiple doses.

## 4. Materials and Methods

### 4.1. Ethics Approval

All animal experiments in this project were performed with animal ethics approval from the Monash Medical Centre Animal Ethics Committee A (MMCA/2015/42, 1 October 2015). Ethical approval for use of human cells in this project was approved by the Monash Health Human Ethics Committee (12387B, 9 April 2013). Experiments were conducted following the Australian National Health and Medical Research Council guidelines.

### 4.2. Cell Preparation

For the collection of umbilical cord tissue, written, informed consent for the use of their placentas was obtained from women that had uncomplicated pregnancies undergoing elective caesarean section at term (<37 weeks).

After clamping of the cord and delivery of the placenta, a 5 cm piece of umbilical cord was collected into a dry, sterile container for processing. Cord tissue samples were processed immediately for isolation of MSCs. Cord tissue was rinsed in phosphate buffered saline (PBS) to remove blood and blood clots. A 1–2 cm section of the cord was isolated and placed in individual 10 cm tissue culture dishes. The tissue was minced manually using a sterile razor blade for a total of 3 min, and then covered in media (16.5% foetal bovine serum (FBS), 1% antibiotics in DMEM:F12; Gibco, Grand Island, NY, USA), and placed in a 5% CO_2_ incubator at 37 °C. Media was changed every 3 days, and cells were passaged when they reached 80% confluency. UC-MSCs were cryopreserved after passage 2 using 10% dimethyl sulfoxide (DMSO) in FBS. To allow recovery of cellular metabolism before administration, seven days before cell infusion, passage 2 cells were rapidly thawed in a 37 °C waterbath and plated in a T175 tissue culture flask (BD Biosciences, San Jose, CA, USA). UC-MSCs were harvested at passage 3, once they reached 80% confluency, for administration to rat pups. At passage 3, UC-MSCs cell surface marker expression was assessed (positive—CD73, CD90, CD105, CD44, HLA-ABC; negative—CD34, CD45, HLA-DR); UC-MSCs demonstrated quad-lineage differentiation (osteoblasts, chondrocytes, myocytes, adipocytes), and cells had a normal karyotype.

### 4.3. Animals

Time-mated Sprague-Dawley rats were obtained from Monash University Animal Research Platform (Clayton, VIC, Australia). All animals were housed at Monash Medical Centre Animal Facility and exposed to standard housing conditions, with 12 h light/dark cycles, with food and water provided ad libitum.

### 4.4. Animal Surgery and Cell Administration

As previously described [26], we adapted the Rice–Vannucci model to induce human term-equivalent perinatal HI, at rat postnatal day (PND) 10 using randomised and coded rat pups (*n* = 34) from 4 litters. Animal codes were maintained throughout analysis to ensure researchers were blinded to the treatment groups. In order to reduce the effect of litter variation, treatment groups were also randomised across every litter, and each litter included sham and HI animals. While sex was not considered in this study, there were similar numbers of females and males in each group.

Throughout the surgery, body temperature was maintained using a heating mat (37 °C). Pups were place under anaesthesia (2% isoflurane, Abbott, Macquarie Park, NSW, Australia), and then a midline incision was made in the neck. Following this, the left common carotid artery was exteriorised and permanently occluded using a cautery device. The incision was then sutured closed, and pups returned to their dam for 1 h. After recovery, pups were placed into a hypoxic chamber for 90 min (BioSpherix, Lacona, NY, USA; 8% oxygen, temperature controlled at 36 °C). The sham group underwent surgery but did not have their carotid artery ligated; they were allowed to recover for 1 h with their dam, then removed for the same duration as the HI animals and kept in normoxic room air on a heating pad at 37 °C. General wellbeing and weight were monitored daily throughout the entire experiment.

On PND11 (24 h post-HI injury) 2 × 10^5^ UC-MSCs were administered via the intranasal route. UC-MSCs were thawed, and cell viability was examined and determined to be >80% for all aliquots. For cell administration, 3 μL of hyaluronidase (100U, Sigma-Aldrich, St. Louis, MO, USA) in phosphate buffered saline (PBS) was first administered to each nostril twice (12 μL in total) to increase nasal mucosa permeability. Thirty minutes later, UC-MSC pups received 2 × 10^5^ UC-MSCs in 12 μL PBS or HI pups received 12 μL of PBS alone, with 3 μL delivered to each nostril twice for both groups.

On PND17, after behavioural testing, animals were culled using an overdose of pentobarbitone sodium (0.1 mg/g). Blood was collected for serum. Brains were collected and weighed before either being fixed in 10% formalin for histology or snap frozen in liquid nitrogen for RNA extraction.

### 4.5. Behavioural Testing: Negative Geotaxis

As previously described [33], on PND 17, the negative geoataxis test was performed by placing the pup head down on a 45° inclined board, to allow pups to move easily move on the board; the surface was covered using a standard laboratory bench pad. The two variables that were recorded included the time it took for the pup to successfully turn 180° (head facing up), and secondly, the time taken for the pup to walk 15 cm up the board to cross a designated line. Each test was repeated three times per pup, for a maximum of 90 s. If pups walked off the board without crossing the line, it was recorded as a fail and not included in the analysis.

### 4.6. Cytokine Analysis

At post-mortem, blood was collected from the heart and placed in Serum Gel-Z tubes (Sarstedt, Numbrecht, Germany), which were then centrifuged at 2000 rpm for 5 min. Serum was collected and stored at −80 °C until used for cytokine analysis. Quantitative analysis of cytokines was performed using a LEGENDplex Rat Th Cytokine Panel (13-plex; Biolegend, San Diego, CA, USA), following the manufacturer’s instructions. Data were acquired using a FACSCanto II flow cytometer (BD Biosciences, San Jose, CA, USA) and analysed using Legendplex software (Biolegend).

### 4.7. Gross Brain Morphology

At post-mortem on PND17, brains were collected immediately after the pups were euthanised. Whole brains were immersion-fixed in formalin for 72 h. Tissue was processed and embedded in paraffin wax, and brain sections were cut at 6 μm for histological analysis. Gross brain morphology and tissue volume were assessed with cresyl-violet and acid-fuchsin stain (Grale Scientific Pty Ltd., Ringwood, VIC, Australia). For each animal, duplicate slides were examined, and data averaged across groups. Images were acquired by Aperio digital scanning (Leica Biosystems, Wetzlar, Germany), and the volume of the left (ipsilateral) and right (contralateral) hemisphere were measured using Aperio ImageScope (Leica Biosystems). For percentage tissue loss, the difference in volume between the contralateral and ipsilateral hemispheres over the contralateral hemisphere volume was calculated, using the following formula ((volume of contralateral − volume of ipsilateral)/volume of contralateral), as previously described [34].

### 4.8. Immunohistological Assessment

Neurons were identified and counted using mouse anti-NeuN antibody (1:200; Chemicon International, Temecula, CA, USA). Microglia were identified using rabbit anti-ionised calcium binding adaptor molecule 1 (Iba-1, 1:500; Wako Pure Chemical Industries, Ltd., Osaka, Japan), raised against a synthetic peptide corresponding to the C-terminal of Iba-1. For morphological assessment of microglia using Iba-1, cells were classified as either resting or activated. Cells with ramified morphology including small cell bodies and multiple long processes were classified as resting microglia, while cells that had enlarged cell bodies with retracted small thick processes, or were amoeboid, were classified as activated microglia. Astrocytes were identified using mouse anti-glial fibrillary acidic protein antibody (GFAP, 1:400; Sigma-Aldrich). All sections were exposed to a secondary antibody (1:200; biotinylated anti-rabbit or anti-mouse; Vector Laboratories, Burlingame, CA, USA) and staining revealed using 3,3-diaminobenzidine (DAB; Pierce Biotechnology, Rockford, IL, USA).

For analysis, images were acquired at 400× magnification under light microscopy (Olympus BX-41, Melbourne, Victoria, Australia); four fields of view per region on two non-adjacent duplicate slides per brain were averaged for each animal. Analysis was performed using Image J (NIH, Bethesda, MD, USA). All assessments were conducted on coded slides and images, with the examiner blinded to experimental groups.

### 4.9. mRNA Expression

The left (injured) hemisphere of the brain was dissected and snap frozen, and RNA was extracted for quantitative real-time PCR. Snap-frozen tissue was homogenised, and total RNA was isolated (Purelink RNA mini kit, Ambion, Life Technologies, Carlsbad, CA, USA) and reverse-transcribed into cDNA (SuperScript III reverse transcriptase, Invitrogen; Life Technologies). Relative mRNA expression was measured by quantitative real-time PCR using the Applied Biosystems 7900HT Fast Real-Time PCR system. The expression of all genes was normalised to the 18S rRNA for each sample using the cycle threshold (Δ*C*_t_) method of analysis and was expressed relative to the sham control group. RT-PCR primer sequences are shown in Table 2.

### 4.10. Statistical Analysis

Results are expressed as the mean ± standard error of the mean (SEM). Statistical analysis was performed using Prism 7.0 (GraphPad Software, version 7.0, San Diego, CA, USA). Experimental and control groups were compared using one-way ANOVA; if significance was identified, post-hoc analysis was performed. For parametric data, Tukey post-hoc analysis was used, and for nonparametric data, Kruskal–Wallis, post-hoc analysis was performed. A value of *p* < 0.05 was considered statistically significant.

## 5. Conclusions

In summary, our results demonstrate that intranasal delivery of a clinically relevant dose of UC-MSCs is an effective therapy for reducing HI-induced neonatal brain injury. Our data suggest that, as with systemic delivery, intranasal administration of UC-MSCs elicits their positive effects predominantly through the modulation of the peripheral and neuroinflammatory response, specifically by decreasing microglia and astrocyte activation. This study provides further evidence for the use of UC-MSCs and cell therapies for the treatment of neonatal brain injury and CP.

## Figures and Tables

**Figure 1 ijms-20-02449-f001:**
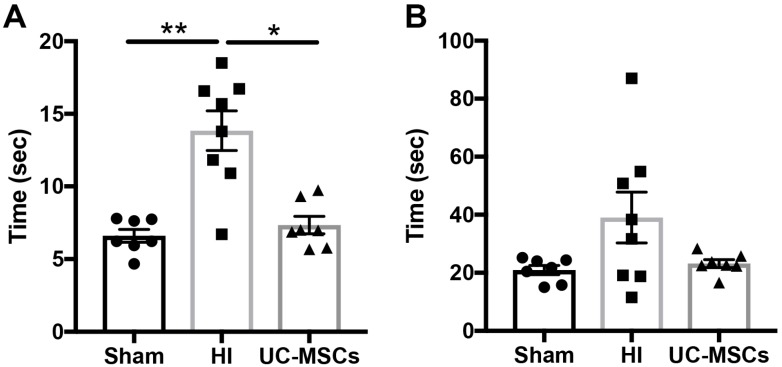
Intranasal delivery of umbilical cord mesenchymal stromal cells (UC-MSCs) improves short-term motor strength and control. On PND14, rats were assessed using a negative geotaxis test. Rats were placed on a 45-degree slope, head down, and the time for them to turn (**A**) and the time for them to walk 15 cm up the board to cross a designated line (**B**) was recorded. (*n* = 7–10 rats per group, * *p* < 0.05, ** *p* < 0.01).

**Figure 2 ijms-20-02449-f002:**
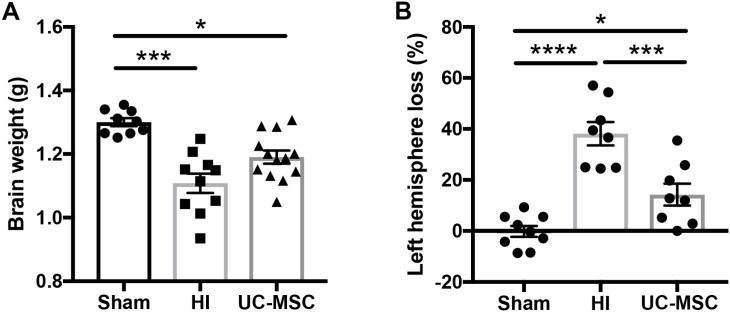
Neonatal hypoxic–ischemic (HI) brain injury results in tissue loss that is improved by intranasal UC-MSCs. (**A**) Brain weight (*n* = 10–13 per group); (**B**) percentage tissue loss of the injured left hemisphere (*n* = 8–9 per group). * *p* < 0.05, *** *p* < 0.001, **** *p* < 0.0001.

**Figure 3 ijms-20-02449-f003:**
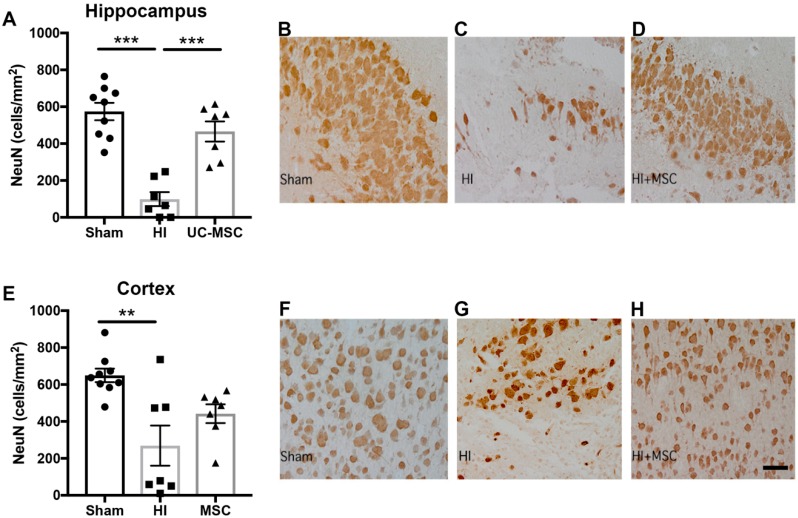
Neuronal cell counts are reduced in the hippocampus following neonatal HI brain injury and improved by UC-MSC treatment. (**A**) Neuronal (NeuN) cell counts in the hippocampus. Representative images of NeuN staining in the hippocampus for (**B**) sham, (**C**) HI and (**D**) UC-MSC treatment; (**E**) neuronal (NeuN) cell counts in the somatosensory cortex. Representative images of NeuN staining in the somatosensory cortex for (**F**) sham, (**G**) HI and (**H**) UC-MSC treatment. All images were taken at 400× magnification, scale bar = 500 μm (*n* = 7–9 rats per group, ** *p* < 0.01, *** *p* < 0.001).

**Figure 4 ijms-20-02449-f004:**
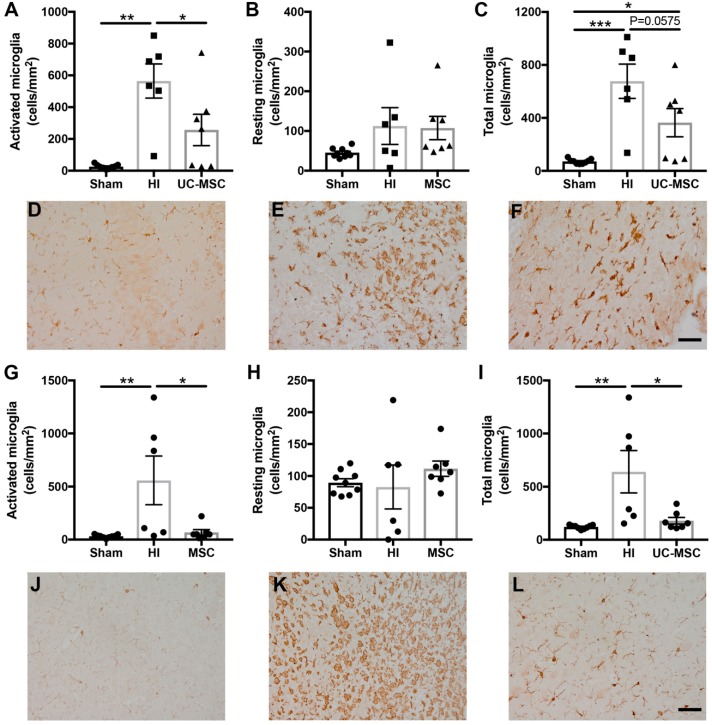
Microglial activation is increased following neonatal HI brain injury and reduced by UC-MSC treatment. Iba-1 immunohistochemistry was performed to identify microglia in the hippocampus, somatosensory cortex and periventricular striatum region of the brain. (**A**) Activated microglia, (**B**) resting microglia and (**C**) total microglia in the hippocampus. Representative images of Iba-1 staining in the hippocampus for (**D**) sham, (**E**) HI and (**F**) MSC treatment; (**G**) Activated microglia, (**H**) resting microglia, and (**I**) total microglia in the somatosensory cortex. Representative images of Iba-1 staining in the somatosensory cortex for (**J**) sham, (**K**) HI and (**L**) MSC treatment. All images were taken at 400× magnification, scale bar = 500 μm (*n* = 6–9 rats per group, * *p* < 0.05, ** *p* < 0.01, *** *p* < 0.001).

**Figure 5 ijms-20-02449-f005:**
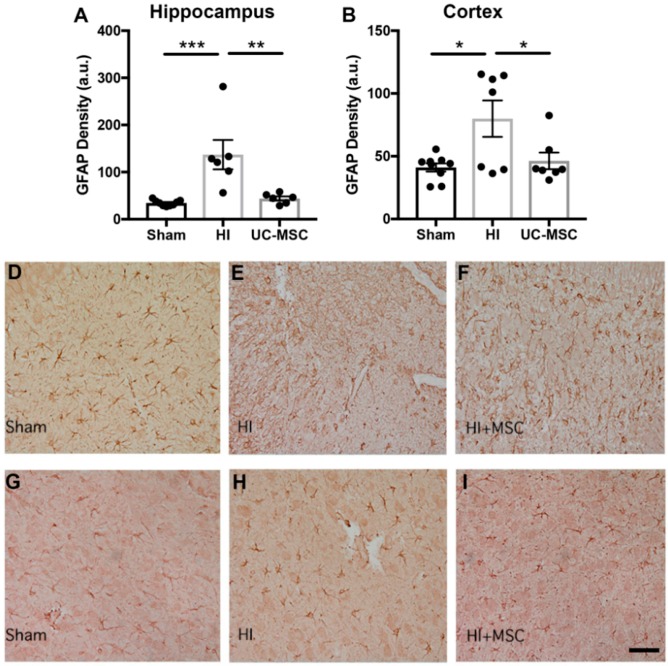
Astrogliosis is increased in the hippocampus after neonatal HI brain injury and reduced by UC-MSC treatment. Glial fibrillary acidic protein (GFAP) immunohistochemistry was performed to assess astrocyte density in the (**A**) hippocampus and (**B**) somatosensory cortex. Representative images of GFAP staining in the hippocampus for (**D**) sham, (**E**) HI and (**F**) MSC treatment. Representative images of GFAP staining in the somatosensory cortex for (**G**) sham, (**H**) HI and (**I**) MSC treatment. All images were taken at 400× magnification, scale bar = 500 μm (a.u.—arbitrary units, *n* = 6–9 rats per group, * *p* < 0.05, ** *p* < 0.01, *** *p* < 0.001).

**Figure 6 ijms-20-02449-f006:**
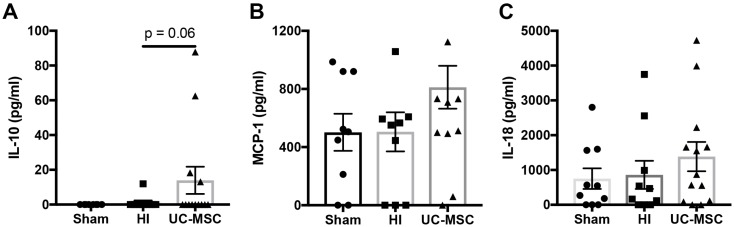
Cytokine analysis of serum following neonatal HI brain injury. (**A**) Interleukin (IL)-10 concentration; (**B**) monocyte chemoattractant protein (MCP)-1 concentration; (**C**) IL-18 concentration. (*n* = 10–13 rats per group).

**Table 1 ijms-20-02449-t001:** Gene expression in injured left hemisphere as measured by RT-PCR.

Gene	Sham	HI	UC-MSC
*BDNF*	1.00 ± 0.07	0.59 ± 0.14 *	0.60 ± 0.04 *
*VEGF*	1.00 ± 0.34	0.39 ± 0.07	0.51 ± 0.07
*IGF-1*	1.00 ± 0.23	16.07 ± 4.72	13.36 ± 8.06
*Claudin 5*	1.00 ± 0.15	0.72 ± 0.13	0.74 ± 0.14
*Occludin*	1.00 ± 0.10	0.76 ± 0.05	1.07 ± 0.30
*GDNF*	1.00 ± 0.22	0.69 ± 0.15	1.47 ± 0.61

Data expressed as fold change from sham (*n* = 5–6 per group, * *p* < 0.05).

**Table 2 ijms-20-02449-t002:** Gene primer sequences used for RT-PCR.

Gene	Sequence	
*VEGFa*	F	AGCGACAAGGCAGACTATTA
	R	AATCCCAGAGCACAGACTCC
*Claudin 5*	F	TTGTGAGGACTTGACCGACC
	R	CTGTTAGCGGCAGTTTGGTG
*Occludin*	F	TATGCTGACCGTAGTACAGAAAGT
	R	TTCCACTCGGGCTCAATCC
*BDNF*	F	AGCAGTCAAGTGCCTTTGGA
	R	CGCTAATACTGTCACACACGC
*GDNF*	F	AAGTTATGGGATGTCGTGGCT
	R	AGAAGCCTCTTACCGGCG
*IGF*	F	CGGGACGTACCAAAATGAGC
	R	CAAGCAGAGTGCCAGGTAGAA

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
