# Peer review of "Intranasal Delivery of Mesenchymal Stromal Cells Protects against Neonatal Hypoxic–Ischemic Brain Injury"

_ijms, 2019, doi:10.3390/ijms20102449_

Round 1
Reviewer 1 Report
The authors have used a rat model of neonatal hypoxic ischemic brain injury to assess the therapeutic potential of a clinically relevant dose of mesenchymal stromal cells sourced from human umbilical cord tissue. The results of this study demonstrated that intranasal administration of 10 million cells/kg at 24 h post-injury was neuroprotective, reducing the loss of brain tissue and neuronal cell loss in hippocampus following injury. Furthermore, treatment with UC-MSC also reduced the injury-induced short-term motor control deficits, significantly diminished the expression levels of neuroinflammation markers as well as degree of reactive gliosis. To my knowledge this is the first time that the consequences of an intranasal UC-MSCs application in context with neonatal brain injury have been investigated. Thus, the results are quite novel. However, there are a number of issues that the authors should address during revision in order to strengthen the impact of this interesting research.
1) In this study, authors used three experimental groups which were mixed across litters to account for litter variation. However, single data for HI group show a broad variance of population distribution than in Sham or UC-MSCs groups. Given the known preclinical and clinical supports of significant sex differences in both the response to and the recovery after brain injury, it would be interesting to know if there are any relations between sex of animals and individual data collected for HI group. What about the differences in weight of animals in the HI group?
2) The distinction of activated from resting microglia based only on the Iba1 immunolabelling is unreliable (especially at later time points after injury). Other markers should be tested and combined with Iba1, e.g. Tmem 119 or CD11b, so that the distinctions in state of microglia as used here could be better and more rigorously confirmed.
3) The authors have examined the changes in degree of reactive gliosis, but they should have also looked at a relationship between UC-MSC treatment and monocyte infiltration. Since the leukocyte influx/monocyte recruitment is known to be an important part of neuroinflammation, the authors should also determine the effect of UC-MSC treatment on monocyte infiltration.
4) Please add the representative images of Iba1 immunostaining in the hippocampus and cerebral cortex for sham, Hi and UC-MSC groups.
5) There are no images of GFAP staining in cerebral cortex shown in figure 5.
Author Response
In this study, authors used three experimental groups which were mixed across litters to account for litter variation. However, single data for HI group show a broad variance of population distribution than in Sham or UC-MSCs groups. Given the known preclinical and clinical supports of significant sex differences in both the response to and the recovery after brain injury, it would be interesting to know if there are any relations between sex of animals and individual data collected for HI group. What about the differences in weight of animals in the HI group?
In this study we were not powered to look at male and female differences as our group numbers were too small, but we did ensure we had similar number of males and females in each group (Sham: 6M/7F, HI (7M/6F), UC-MSC (6M/7F). We have added a line to the methods on page 9 to reflect this. In another unpublished study using this same model but assessing the long term outcomes, we have analysed sex differences and have not found any significant differences in neurotopathological outcomes between males and females in our model.
As for variation in the HI group, there were no significant differences in the body weights at PM between any groups; sham 30.1 ±1.9g, HI 26.7 ±2.3g, UC-MSC 30.7 ±1.4g and the variation in the body weight of the HI group was not larger than the two other groups. So we do not belive that body weight was the source of the variability. This model is known to have a high degree of variation, however the source of this is not always known. We have added the body weights of each group into the results text (page 3) for further information.
The distinction of activated from resting microglia based only on the Iba1 immunolabelling is unreliable (especially at later time points after injury). Other markers should be tested and combined with Iba1, e.g. Tmem 119 or CD11b, so that the distinctions in state of microglia as used here could be better and more rigorously confirmed.
We agree with the reviewer that assessing activation of microglia primarily based on morphology is not the most reliable method for distinction, though it is a very commonly published method in the perinatal field and in rat studies, which have limited antibodies available compared to mice. The challenge with defining microglial activation with antibodies is also difficult and there is much debate as to what are the most reliable markers for eg M1 and M2 microglia. While Tmem119 and CD11b may be better at identifying microglia compared to macrophages, Tmem119 is hard to get working in the rat, but neither really identify the difference between different microglial activation states. We have previously tried double labelling cells with Iba-1 and CD68 (for M1 pro-injury microglia), and Iba-1 and mannose receptor (M2 pro-reparative microglia), but we found this to be unreliable and not easy to replicate. This is the reason we have chosen to use Iba1 and and morphology assessment to give us an indication of microglial activity. We have added text to the discussion to highlight this as a potential limitation in our study (page 8).
The authors have examined the changes in degree of reactive gliosis, but they should have also looked at a relationship between UC-MSC treatment and monocyte infiltration. Since the leukocyte influx/monocyte recruitment is known to be an important part of neuroinflammation, the authors should also determine the effect of UC-MSC treatment on monocyte infiltration.
We agree that the systemic recruitment of immune cells to the CNS is very important to injury. We have previously published that in this neonatal model of HI brain injury, there is a significant increase in Th1 T cell recruitment into the brain at 7 days post injury (McDonald 2018, J Neuroinflam, 15;47). In addition we have shown that treatment with different cell types found in UCB are able to reduce this recruitment and we proposed this is one mechanism of action of cell therapy (McDonald 2018, J Neuroinflam, 15;47). While that study did not use the same cells that were used in this current study (UCB vs UC-MSC), previous papers have shown that UC-MSCs and UCB have similar anti-inflammatory actions.
Of relevance to the actions of UC-MSCs, we have recently published work in a sheep model of preterm brain injury that demonstrates that UC-MSC and UCB treatment does reduce monocyte recruitment to the brain using MPO (Paton 2018, Dev Neurosci, 40;258; Paton 2019, Ped Res,doi: 10.1038/s41390-019-0366-z).
We agree that this information is necessary, but to obtain this information we would need to complete a different set of animals to investigate these mechanisms, therefore it would be a future study that is very important. We have included this information and future directions to the discussion (page 8).
Please add the representative images of Iba1 immunostaining in the hippocampus and cerebral cortex for sham, Hi and UC-MSC groups.
We have added representative images of Iba-1 staining in the hippocampus and cortex to Figure 4.
There are no images of GFAP staining in cerebral cortex shown in figure 5.
We have added images of GFAP staining in the cortex to Figure 5.
Reviewer 2 Report
Dear colleagues!
After assessment of a manuscript by McDonald et al. as a Reviewer assigned by Edtiorial board I have the following comments on the study submitted to IJMS:
Overall, this work presents a novel approach in cell therapy of high interest and definitely has high scientific merit and interest for the Reader.
However, to strengthen the study further improvements might be required and I suggest the Authors to comment on the following issues:
1) The Reader might be interested regarding the cell fate - whether they retain, undergo apoptosis and for how long might they remain in site of delivery to render effects; Please, comment on that or provide additional experimental data to clarify the point.
2) Tests in figures 1A and 1B should be commented in Discussion in more detail as far as it seems that "designated line" test failed to show impairment in HI group. Please, comment on that or (which could be the best) - provide a supplementary video that would be an excellent convincing element overwhelming any "p-values".
3) Study limitations might be stated in the manuscript, namely:
a) lack of long-term outcome in treated animals showing that therapy prevented development of deficits and impairments in later terms (using behaviour tests etc)
b) Potential immune response to human UC-MSC - please comment on that point and I suggest to comment on a potential link between serum ELISA suggesting anti- inflammatory IL-10 (overall, I suppose that endpoint analysis was not the best choice which potentially resulted in in-significant results of IL-8 and MCP-1.
Finally, please comment on the protocol using fresh-thawed UC-MSC. This kind of preparation has been reported by J. Galipeau and K. Leblanc as not the best option for "paracrine-mediated" cell therapies, which is definitely the case described in this work (https://www.tandfonline.com/doi/full/10.3109/14653249.2011.623691).
Please, comment on lack of at least short-term "recovery" period in vitro for MSC, which has been reported to "revive" MSC pararcrine and immunomodulating action. Furthermore, the latter is of significant importance in a xenogeneic model used by Authors.
Regards, Reviewer.
Author Response
Reviewer 2 comments
1)The Reader might be interested regarding the cell fate - whether they retain, undergo apoptosis and for how long might they remain in site of delivery to render effects; Please, comment on that or provide additional experimental data to clarify the point.
As part of study we did actually label a group of UC-MSCs with CFSE (Carboxyfluorescein succinimidyl ester)to try and determine the fate of these cells after intranasal delivery. When we assessed the brains 7 days later we were not able to find any cells in the brain. We weren’t surprised by this finding though, as it is a common understanding that cell engraftment is not necessary for therapeutic benefit of mesenchymal stromal cells, therefore we have not included this in our study. We have added a sentence to the study summary at the start of the discussion to state we did not find cells within the brain (page 7).
2) Tests in figures 1A and 1B should be commented in Discussion in more detail as far as it seems that "designated line" test failed to show impairment in HI group. Please, comment on that or (which could be the best) - provide a supplementary video that would be an excellent convincing element overwhelming any "p-values".
We have added more detail about these results in the discussion on page 8.
3) Study limitations might be stated in the manuscript, namely:
a) lack of long-term outcome in treated animals showing that therapy prevented development of deficits and impairments in later terms (using behaviour tests etc)
We agree that the long term outcomes are very important, especially since we have just published that assessment of short term efficacy doesn’t always reflect long term improvements (Penny, Frontiers Physiology, 2019), we have now added more detail about this test in the discussion (page 8).
b) Potential immune response to human UC-MSC - please comment on that point and I suggest to comment on a potential link between serum ELISA suggesting anti- inflammatory IL-10 (overall, I suppose that endpoint analysis was not the best choice which potentially resulted in in-significant results of IL-8 and MCP-1.
We cannot rule out the potential that there may have been a xenogeneic response to the human cells put in our rat model, however as these are neonatal rats they have a more naïve immune system than adults and therefore it may be that they have a reduced ability to mount these types of responses. We are currently trying to investigate these exact questions, but currently do not have the answer.
As for the cytokine analysis, we are also hampered by the fact that cytokine responses in neonates are dulled compared to adults and while we tested 13 different cytokines (IL-10, IFN-y, CXCL1, MCP-1, TNF, GM-CSF, IL-18, IL-12p70, IL-1b, IL-17A, IL-33, IL-1a, IL-6) the only ones we detected were IL-10, IL-18 and MCP-1, the others were all undetected. That is the reason we only reported on those three. We have previously used plasma and this has not improved detection. It would also be good if we could assess cytokines in CSF as this is the site of injury, however to obtain CSF from rat paps is extremely difficult.
Finally, please comment on the protocol using fresh-thawed UC-MSC. This kind of preparation has been reported by J. Galipeau and K. Leblanc as not the best option for "paracrine-mediated" cell therapies, which is definitely the case described in this work Please, comment on lack of at least short-term "recovery" period in vitro for MSC, which has been reported to "revive" MSC pararcrine and immunomodulating action.
We apologise for the confusion in our methods. Our cells were actually thawed (passage two cryopreserved cells) and cultured to 80% confluence before they were harvested and administered (at end passage3) to the rats. We agree that allowing the metabolic recovery of the cells from cryopreservation is important. We have reworded the methods to clarify this point (page 10).
Round 2
Reviewer 1 Report
The authors have adequately attended to the suggested revisions and have addressed the issues raised by the reviewers. However, a number of details are missing from the revised manuscript, so that some revisions in the text and figures are still needed.
1) The conclusion that there are “no significant differences in neuronal cell number between the three groups in the somatosensory cortex” (line 117-118) is not supported by the results of quantification in Fig. 3E indicating that neuronal cell counts in the somatosensory cortex are significantly reduced (**P<0.01!!!) following neonatal HI brain injury. Such a marked loss of neurons in the HI-affected cortical parenchyma is however not visible in Fig. 3G. Therefore, the example for NeuN staining in Fig.3 should be replaced by a truly representative image that has strong validity in relationship to the observed changes in this brain region.
2) The authors must give many more details of the classification microglial activation based on immunolabelling for Iba1 to convince the readers (especially ones not familiar with the perinatal field) that UC-MSC treatment indeed effects the state of microglia in different regions of the neonatal brain. It is therefore necessary to know which morphological parameters (i.e. cell area, volume, perimeter, branch complexity, lacunarity etc.) were exactly selected for the categorization of microglia state. Additional information about the value of selected parameters in the three experimental groups should be supplied either in the text or in figure/table, and would allow to understand how microglial cells were thus classified as activated or resting ones.
3) I did not find any scale bars in the revised Figures 3-5. Please add the valid scale bars to photomicrographs and define them in the figure legends.
Author Response
1) NeuN in somatosensory cortex description in text and replace representative image Fig 3G.
We apologise for this error in our description of the results. We have now amended the text and replaced Fig 3G with a more representative image.
2) The authors must give many more details of the classification microglial activation
This was an oversight for not including the description for how we characterised microglia in our methods section. This information has now been included on page 11 lines 396-400. We hope that this improves the clarity of how we classified our microglia.
3) I did not find any scale bars in the revised Figures 3-5. Please add the valid scale bars to photomicrographs and define them in the figure legends.
Scale bars have now been added to each figure (Fig 3-5) and described in the figure legends.